# A Nested Ensemble Approach with ANNs to Investigate the Effect of Socioeconomic Attributes on Active Commuting of University Students

**DOI:** 10.3390/ijerph17103549

**Published:** 2020-05-19

**Authors:** Khaled Assi, Uneb Gazder, Ibrahim Al-Sghan, Imran Reza, Abdullah Almubarak

**Affiliations:** 1Department of Civil and Environmental Engineering, King Fahd University of Petroleum & Minerals, Dhahran 31261, Saudi Arabia; ialsghan@kfupm.edu.sa (I.A.-S.); ireza@kfupm.edu.sa (I.R.); 2Department of Civil Engineering, University of Bahrain (Issa Town Campus), Zallaq 32038, Bahrain; unebgazdar@gmail.com; 3Department of Civil and Environmental Engineering, UC Berkeley, Berkeley, CA 94720, USA; mubarakafm@berkeley.edu

**Keywords:** nested ensemble ANN, mode choice, active commuting, sensitivity analysis, university students

## Abstract

Analysis of travel mode choice is vital in policymaking and transportation planning to comprehend and forecast travel demands. Universities resemble major trip attraction hubs, with many students and faculty members living on campus or nearby. This study aims to investigate the effects of socioeconomic characteristics on the travel mode choice of university students. A nested ensemble approach with artificial neural networks (ANNs) was used to model the mode choice behavior. It was found that students generally prefer motorized modes (bus and car). A more detailed analysis revealed that teenage students (aged 17–19 years) had an approximately equal probability of selecting motorized and non-motorized modes. Graduate students revealed a higher tendency to select motorized modes compared with other students. The findings of this study demonstrate the need to promote non-motorized modes of transport among students, which is possible by providing favorable infrastructure for these modes.

## 1. Introduction

The assessment of transport mode choice significantly contributes to city planning and policymaking by predicting trips generated in the medium of interest [1]. Saudi Arabia is plagued with poorly designed metropolitan areas that do not offer adequate public transportation or an integrative network for pedestrians and bicycles. Diversified forms of transportation for everyday commuting are not common, as most Saudi citizens rely on automobiles in their daily commute. For instance, Jeddah city has seen a drastic increase in daily automobile trips, from 50% in 1970 to 93% in 2007 [2]. Additionally, mode selections for students in public schools in Al-Khobar city, Saudi Arabia, are limited to the use of personal passenger cars and walking in hindered conditions [3].

Current Saudi youth who live in major cities have been exposed to urban surroundings characterized by high dependency on automobiles as the primary form of transportation. Although citizens have a positive attitude toward using active forms of transportation, such as bicycling, they abstain from commuting via bicycle due to unsafe conditions and a lack of sufficient infrastructure in Jeddah city, Saudi Arabia [2]. After decades of subsidized gasoline prices, such prices increased from 0.24 USD/liter to 0.54 USD/liter in 2018, which might have caused the Saudi youth population to reconsider its transport modal choices, as this was previously observed in different developed countries [4]. According to a study in Germany, young adults lowered their reliance on automobiles compared with the significant increase in automobile trips in the 1990s. This reduction in preference for automobiles has been attributed to changes in fuel prices, urban density, and communication technologies that substantially impact commuters’ modal choices [5].

Recently, university students’ modal choice patterns have been studied more intensively. It is predicted that they are more inclined to adopt active modes compared with other population classes [6]. Studies on university students’ mode choice in Vietnam indicated that time of travel, school location, socioeconomic factors, bicycle ownership, automobile ownership, and car lane separation on roads are critical factors that alter preferable modal decisions [7]. Furthermore, seasonable variations were noticed to impact commuting patterns in universities in the United States [8]. However, studies in Iran revealed that university students prefer using passenger cars over walking and cycling [9].

Understanding the concept of sustainable transportation is essential to ensure a clean, healthy, and high-quality environment. The sustainable transportation concept also emphasizes human life and the environment to meet current and future needs, without considering the ability of the future generation to meet its own needs. Such a promising concept of sustainable transport for consumers ensures the safety of road users and brings about social welfare to society and, thus, to the living environment. There are several significant sustainability impacts of transportation facilities and activities, some of which are listed in Table 1.

Active commuting by university students might appear to have limited effects on society, as it occurs only at the university campus. However, it could be considered a stepping stone for achieving a sustainable society. Only a limited number of studies have been conducted on university populations, not including any travel behavior study on a university population in the context of Saudi Arabia. Thus, this study aims to explore the effect of socioeconomic attributes on active traveling at the university level. A nested ensemble approach with artificial neural networks (ANNs) was used to assess the effect of socioeconomic factors on students’ active commuting decisions. The uniqueness of the ANNs-based modeling approach used for this research is unprecedented in this field of study. It is expected that the results of the study can be utilized to identify the influencing factors for active commuting for university populations throughout Saudi Arabia. These factors can be considered useful for regional travel demand modeling. Furthermore, this study utilizes a rich dataset on the activity-travel behavior of a large Saudi Arabian university population, which can be considered a valuable addition to the literature. Knowledge of critical factors influencing the choice for such active modes might serve as a valuable input parameter for university policymakers.

The paper is organized as follows: Section 2 contains a review of the relevant literature. Section 3 describes the study area. The data collection method used in the study is then described in Section 3, after which the methodology is presented in Section 4. Results are presented and discussed in Section 5. Finally, Section 6 outlines the main conclusions and identifies and recommendations for further research.

## 2. Literature Review

Universities worldwide are not only large employers, educational institutions, and pioneers in community leadership but also large trip generators and attractors [7]. University planners and management authorities are putting greater focus on environment-friendly and non-combustion-based modes of transport due to the rise in climate change awareness, higher on-campus parking demand, hike in gasoline prices, and added taxes. Such modes, often referred to as sustainable transport, include all active commuting in the forms of public transport (e.g., buses and mass transit, among others) and non-motorized modes of transport (e.g., walking and bicycling, among others).

University students represent a social group that tends to have complex and diverse travel behavior [11]. Unlike others, university students are more open to adapting and trying new ideas from friends and colleagues of mixed interests and diverse cultures. The changing nature of the class schedule allows these students to enjoy various activities throughout the day. All such factors contribute to creating convoluted daily schedules of university students, resulting in complex and uncertain travel patterns.

Walking, cycling, and public transport use are treated as active modes of transport due to the involvement of physical activity. Even public transport is classified into the “active mode” category, as it involves walking or cycling at the beginning or end of the journey. Therefore, planning strategies that inspire students to use an active mode of transport would directly impact reduced on-campus parking demand and would thus help the university’s environment to become cleaner due to lesser emission. At the same time, these strategies would benefit students with improved health, as various studies on adolescents reveal that increasing physical activity is inversely proportional to depression and positively influences academic performance [12,13], thereby highlighting another potential gain for university students who actively commute. Another study found that students who regularly use bicycles or public transport have increased potential to achieve satisfactory physical activity levels to meet public health recommendations than do students who travel by private vehicles [14,15].

The mode choices of university students could be affected by the local environment and policy actions. A study found that students’ mode choices at the University of North Carolina in Chapel Hill were significantly affected by the local environment, which includes the topography and the availability of pedestrian facilities [16]. Another study by Balsas [17] focused on devising policy actions to promote sustainable transport systems at eight American universities. This study revealed that such policy actions produced significant changes in commuters’ modes of travel. Campuses with well-structured bicycling and walking facilities influence commuters to shift their mode of transport from motorized to active commuting. Akar et al. [18] estimated mode choices for home-to-campus trips based on a travel survey conducted at Ohio State University in 2011. The results revealed that the students are more likely to use bikes and transit modes for home-to-campus trips compared with faculty members and staff. Another school-day activity–travel diary survey conducted at North Carolina State University revealed that students living on campus have a higher number of out-of-home activities than do those living off-campus [19]. Similarly, the results of a one-week travel diary survey conducted at a rural university in Thailand indicated high social interdependency among some pre-selected student groups that govern their travel patterns [11]. Moreover, an online-based travel diary survey conducted at the University of Western Australia examined the commuting patterns, barriers, and motivators that affect travel decisions of university populations. They found that removing barriers to using active modes significantly impacts commuting patterns [7]. In 2009, a travel diary survey was conducted for four American universities for use in conjunction with travel behavior data gathered from the National Household Travel Survey (NHTS) to identify the difference of travel patterns between university students and the general population. Based on the analysis, the study placed the students in a relatively lower income group with atypical travel behavior [20].

In most cases, travel demand models use socioeconomic factors (e.g., household size, household income, and car ownership) to consider trip rates generated for the general population, which might not resemble the travel behavior of the university population [21]. From a broader perspective, university campuses offer more opportunities to participate in different types of activities within a reachable distance because of their mixed land uses with lively characteristics of the built environment. These unique features cause a distinction of university travel behavior from the general population and, thus, require travel-demand-specific management strategies [22].

The case study by Zhou [23] re-confirms that university students are likelier to share a residence based on factors such as rent affordability, bus proximity, and shorter travel. Students’ choice of alternative transport is significantly influenced by transit pass subsidies. In another study, frequent bus schedules, “free-rider” public bus passes, and dedicated city bike lanes were found to influence the university user groups (students, faculty, and administrative staff) to select alternative transport [24]. Miralles-Guasch et al. [25] found a higher inclination toward the use of active travel modes by millennial university students at an autonomous university metropolitan campus in Barcelona (Spain). The strong bias toward an active travel mode by the university students could be due to several factors, such as active transportation infrastructures, willingness to use alternative travel modes, psychosocial benefits related to safety, economic issues, environmental concerns, and influences of the latest information and communication technologies [26].

Policies conducive to non-motorized and public transportation can immediately benefit the local environment in terms of reduced air pollution, parking areas, and traffic congestion. However, in the long run, the effect is more promising, as students’ adaptation to reduced automobile dependency would remain in their adult and professional lives and might be reflected in policy when some end up in careers related to policymaking [7]. As habit governs the decision to select non-motorized transportation, such as bicycling, over cars [27], shaping these habits at the onset of adulthood could significantly impact the community and the overall transportation system.

While researchers have reported the use of parametric and non-parametric statistical models in analyzing mode choice patterns of university students, the use of artificial intelligence in full scale from this perspective is infrequent. Akar et al. [18] used a discrete choice model to analyze the mode choice of pupils at Ohio State University, while few others have used multinomial logistic regression, nested logit models, and tree-based models in their analyses [16,28,29,30,31]. Bicikova [32] used a two-step cluster analysis to segment and profile British students’ travel based on their motivational and behavioral characteristics. Ermagun et al. [33] used a data-mining method, a random forest method, and an econometric (nested logit) approach to perform a comparative analysis to identify a connection between mode choice and escorting decisions for school trips. They estimated that the random forest method’s prediction potential was almost double that of the nested logit model. In another study, Daisy et al. [22] used zero-inflated negative binomial (ZINB) models to identify the relationship between university students’ tenure of housing and choice of sustainable transport. Assi et al. [34] reported the use of three machine learning tools (MLT), namely extreme learning machine (ELM), support vector machine (SVM), and multi-layer perception neural network (MLP-NN), to predict school pupil mode choice to school. One major drawback of this study is that it had only a binary output (i.e., students either choose to travel by car or to walk). However, the successful implementation of MLTs, as suggested by the results of this study, is taken as a motivation to use MLTs for university goers’ mode choice with more input variables in this study. Each model has its own set of advantages over others and is also used based on the type of problems presented. For example, in discrete choice models, the number of alternatives is confined. However, in the ordinary regression model, the dependent variable can take an infinite number of values. Multinomial logit (ML) relies on the assumption of independence of irrelevant alternatives (IIA), which is not always desirable. For example, in the ML model the relative probabilities of taking a car or bus to university do not change if a bicycle or other mode of transportation is added as an additional possibility. Thus, often times ML models impose too many constraints on the relative preferences between the different alternatives. Nested logit or the multinomial probit model could be used to solve this problem, as they allow for violation of the IIA. With all these compulsions in mind we found that the juxtaposed use of MLTs along with nested ensemble has not been successfully used for mode choice problems in literature, which has resulted in its non-utilization in mode choice problems with a pool of similar choices. Hence, this study attempts to fill this gap.

## 3. Study Area

King Fahd University of Petroleum and Minerals (KFUPM) is located in Dhahran, a city on the eastern coast of Saudi Arabia, which is home to around 10,000 aspiring students. High temperatures and high humidity levels characterize Dhahran’s climate during the summer—which can reach 46 °C (114.8 °F)—while moderate weather characterizes the rest of the year [35]. It should be mentioned that KFUPM provides subsidized dorms for all students; thus, most of the students live on campus. The map of the study area is shown in Figure 1. The student housing zone (gray building) is located on the north side of the campus, and academic buildings (blue buildings) are located on the west and south side of the campus.

Furthermore, students commute to classes via buses, passenger cars, walking, and bicycling. KFUPM operates a free bus service, which transports students from the student dorms to academic buildings on campus. The bus system is composed of three bus lanes that are scheduled for pickups on an average of every five minutes.

Moreover, KFUPM provides vast parking areas free of charge for students living outside the campus, where parking on campus has not been a problem at KFUPM. Walking on campus is facilitated through somewhat wide sidewalks and crosswalks. Additionally, several students choose to commute to classes via bicycle; however, bicycle infrastructure has not yet been provided at KFUPM, which threatens the safety of bicyclists commuting to classes.

## 4. Data Collection and Description

A questionnaire was designed based on various factors that were noted to impact the modal choice, as indicated in the literature review. Information about present mode choice behavior, age, college level, monthly income, and car ownership was collected to provide the basis for the analysis. The study was limited to these variables since their relation to mode choice has been observed in the literature [8,22,31,36]. A participatory pilot study was conducted before the full-scale survey to ensure that the questions and answers of the questionnaire were meaningful [37]. The questionnaire was administered to a small sample of students. The students were interviewed and told that they are in the pre-test phase. Then, they were asked about their comments and suggestions on the questionnaire. All issues discovered during the pilot survey phase were addressed. The responses to the questionnaire were interpolated to develop a modified questionnaire that was used for this study. The full-scale survey was administered via an online survey through KFUPM’s student affairs office, online forums, and Twitter hashtag. The total number of responses was 246, of which only 229 were considered valid for this study, which represents an acceptable sample size for study areas with few subgroups such as KFUPM [38,39]. A detailed description of the variables and their use in the model are provided in Table 2.

Based on the descriptive statistics of the valid questionnaires, it was found that 48% of students were currently commuting by bus, 13% by car, 13% by bicycle, and the rest (26%) via walking. Moreover, the college level split was noted as follows: 17% of the respondents were in their orientation year, 49% were freshmen, 16% were sophomores, 6% were juniors, 9% were seniors, and the remaining 3% were graduate students. Furthermore, the respondents’ ages ranged from 17 to 29 years of age, with an average respondent age of 20.2 years and a standard deviation of 1.6 years. Respondents’ average monthly income was found to be SAR 1400 ($374), with a standard deviation of SAR 800 ($213). Finally, 69% of the respondents owned an automobile. Information regarding the 229 valid response samples is summed up in Table A1 in Appendix A. An artificial neural network (ANN) was used in this study because it does not have restrictions of statistical significance or the requirement of knowing the exact relationships between input and output variables. It would have been challenging to use traditional statistical models (such as logit and probit, among others) on these data with a small number of variables because of these restrictions.

## 5. Modeling Methodology

Students’ mode choice was predicted using a series of ANN models, depicted in Figure 1. For this approach, the final mode choice is determined in two steps; the ANN (top nest) was to predict the type of mode (motorized and non-motorized), while the actual mode choice was predicted in the last step, with another ANN, depending on the category of mode (motorized or non-motorized) the traveler selected in the first step. Therefore, each classification of mode choice is referred to as a nest.Since the final mode choice is an outcome of two ANNs, it is referred to as an ensemble. The same is illustrated in Figure 1, wherein the top nest of the ensemble predicts the choice of students for the motorized or non-motorized mode of transport. The former nest includes car and bus, while the latter includes walking and bike. This segregation was performed on the basis that walking and bike riding are considered active modes of transportation while others involve a motorized vehicle. The second layer of the ensemble predicts the final choice of the students. This approach provides a more detailed description of the travelers’ choice and their behavior by dividing it into two steps: motorized and non-motorized layers, as shown in Figure 2.

Typical feedforward ANNs [40] were used to model the top nest and the final choice. A simple structure with three layers (input, hidden, and output) was assumed sufficient for this study, as the number of samples and variables was not very high. The typical network is presented in Figure 3. The hyperbolic activation function was used for the neurons in the hidden layer.

All independent variables were normalized using a min − max function to reduce the noise in the data, as per Equation (1) [41]:(1)$$x^{\prime }=\frac{x-Min\left(x\right)}{Min\left(x\right)-Max\left(x\right)}$$
where x′ is the normalized value of x. We acknowledge that numerous possible combinations can exist for ANNs and that testing all of them would not be possible for such a study. Thus, the above parameters were kept constant. Moreover, the number of neurons in the hidden layer was changed to maximize the accuracy within this architecture. Different cases of neurons in the hidden layer were used, and the accuracy of the ANN was evaluated. Finally, the number of neurons with higher accuracy was retained as the final network.

The complete process of modeling and analysis is shown in Figure 4. It starts with the distribution of data into different nests according to the mode choice of travelers (see Figure 2). Then each dataset was divided, through randomly sampling, into training and testing datasets with 75% samples for training and 25% for testing. The dataset consisted of 229 samples. Thus, the training dataset was 172 samples, while the test dataset was 57 samples for the top nest ANN model. There were 141 samples for the motorized mode choices, among which 106 were training samples, and 35 were test samples for the ANN model developed for motorized mode choice nest. Lastly, 88 samples were available for non-motorized mode choice nest, and 66 were for training, and 22 were the test of ANN model for non-motorized mode choices. ANNs were developed using Statistica (by Statsoft Inc., Hamburg, Germany) Then, their accuracies were compared to find the optimum number of neurons. Finally, a sensitivity analysis was performed by selecting one variable at a time and changing its value from the given range in the data sample. The model was used to predict the values of the mode choice of students for each ANN by changing the value of one variable while other variables were not changed in that case. For example, to perform sensitivity analysis for age, only its value was changed while other values were constant. The probabilities of mode choices, predicted by the model with the change in the values of a variable, were then plotted (see Figure 5, Figure 6, Figure 7, Figure 8, Figure 9, Figure 10, Figure 11, Figure 12, Figure 13, Figure 14, Figure 15 and Figure 16) to show the effect of that variable on them. In each case, the model was used to perform a sensitivity analysis by changing the values of specific variables while keeping other variables at a constant value (average of the variable). The range of parameters was taken from the actual data; for example, the income of students ranges from approximately SAR 800 ($213) to SAR 7000 ($1867), as it was the range of income for the top nest (see Figure 7). On the other hand, students using non-motorized modes had a lower income range (SAR 840–SAR 2590).

## 6. Results and Discussion

As previously stated, the modeling was performed in two steps using ANNs as nested ensembles. The following sections provide the results and elaboration for these steps. The accuracies of all ANNs are presented in Table 3. Ten hidden neurons were found to have the highest possible accuracy on test datasets for all models, which could be because the number of neurons depends on the type of problem and independent variables that were the same in all cases. The accuracy calculated for the test dataset was approximately 60% for all ANNs. A higher accuracy could be achieved by incorporating trip information, which was not the focus of this study. Instead, it solely concerns the effects of socioeconomic characteristics of the students on their mode choice behavior. However, the models developed in this study could be considered satisfactory (with this accuracy level) for determining the relationships between the mode choice and socioeconomic characteristics of students.

### 6.1. Top Nest ANN

First, ANN was developed to determine the mode choice of students between motorized and non-motorized modes. A sensitivity analysis was conducted for each variable using the top nest ANN, as displayed in Figure 5, Figure 6, Figure 7 and Figure 8. As shown in Figure 5, teenagers (17–19 years) had an approximately equal distribution of mode choice for motorized and non-motorized transport. The choice of motorized transport increased with age, which can be explained by the fact that older students might be more familiar with the bus routes and parking areas and might prefer buses and cars over walking or bicycles.

Figure 6 and Figure 7 reveal that a general preference existed for motorized modes by the students, regardless of car ownership or income. The preference for motorized modes could be due to the hot weather conditions in Saudi Arabia, which remain as such throughout the day during most times of the year. However, it could be seen that the probability of motorized travel further increased with car ownership and an increase in income.

Figure 8 illustrates that the probability of selecting a non-motorized mode of transport decreased for senior students and especially graduate students. It should be noted at this point that many of these graduate students either had a job or were on-scholarship; therefore, they would find it easier to own and maintain a car. It is consistent with the observations from Figure 5.

### 6.2. Motorized Nest ANN

The sensitivity analyses of the ANNs developed to predict the choice of students among motorized modes of transport (bus and car) are presented in Figure 9, Figure 10, Figure 11 and Figure 12. It could be commonly observed from these figures that the use of the bus was generally preferred in most of the cases. Contributing factors for this phenomenon could be shorter trips, the available frequency of buses, and avoiding the search for parking spaces. However, age negatively impacted the probability of bus use, as shown in Figure 9. On the other hand, car ownership positively influenced car use probability, as presented in Figure 10.

The probability of choosing the car mode was found to be higher than that for the bus for students with higher income levels and in their sixth year of education (see Figure 11 and Figure 12). The possible explanations for this trend could be that graduate classes were held in the evening, at which time the buses did not operate, and parking spaces were mostly empty. Second, some of these students using motorized modes could have been working outside (also indicated by their higher income). Thus, they might have been coming directly from their workplaces. In this case, the optimum income level was SAR 6500 ($1733), at or above which the students shifted from bus to car, as depicted in Figure 11.

### 6.3. Non-Motorized Nest ANN

A second nest ANN was developed to predict the mode choice of students among non-motorized modes (i.e., walking and bike). The sensitivity analysis for this model is displayed in Figure 13, Figure 14, Figure 15 and Figure 16. Students who opted for non-motorized modes generally preferred walking over biking, as illustrated in the figures. Figure 13 reveals that students had an equal liking for biking and walking at 22 years of age, while the probability of biking was higher for older students.

Car ownership did not appear to significantly impact the choice among non-motorized modes, as presented in Figure 14. Figure 15 reveals that students with an income of approximately SAR 2100 ($560) had an approximately equal probability of choosing biking and walking, while students who had an income higher than that tended to have a higher preference for biking. The range of students’ income showed in Figure 15 is lower than the range of the students’ income showed in Figure 7, as it is only for non-motorized modes, which would not be preferred by students with higher income levels. Graduate students who chose non-motorized modes had an approximately equal proportion of selecting walking and biking, as displayed in Figure 16.

## 7. Conclusions 

Universities resemble major trip attraction hubs, with many students and faculty members living on campus or nearby. This study aimed to explore the effects of socioeconomic characteristics on the mode choice of students at KFUPM. A nested ensemble approach with ANNs was used in this study to model the mode choice behavior. An online questionnaire was designed based on factors that were noted to impact the modal choice, as indicated in the literature review. Information about the present mode choice behavior, age, college level, monthly income, and car ownership was collected to provide the basis for the analysis.

It was found that students generally prefer motorized modes (bus and car). Teenage students (17–19 years) had an approximately equal probability of selecting motorized and non-motorized modes. Graduate students demonstrated a higher tendency to select motorized modes compared with others. The bus was preferred among motorized modes, wherein cars were preferred only by graduate students with income more than SAR 6500 ($1733). Walking was more prevalent among students who prefer non-motorized modes, and biking was more commonly used by graduate students with income higher than SAR 2100 ($560).

The study’s findings reveal that motorized modes (especially cars) were more preferred by the students. Hence, there is a need to take measures to promote non-motorized modes of transport among young students, especially those with higher income levels. It has been found from the review of literature that this could be accomplished by providing favorable infrastructure for these active modes; this includes providing shaded pathways, bike lanes, and parking spaces. Moreover, providing insight into the benefits of commuting via bicycles to orientation students and first-year students could motivate them to use bikes at an early stage of their college career. Operating a shuttle service in the evenings and connecting it to nearby off-campus locations also might encourage graduate students using motorized modes, who are presently biased to commute by cars, to use the service. However, the effects of these measures could be studied in detail if and when they are implemented. Further research could be done to include the effects of trip data (such as cost, time, distance, and schedule, among others) on the mode choice of students, which was not the focus of this study. Another avenue of research could be to determine the effects of possible interventions (through any/some of the measures as mentioned earlier) through stated choice data.

## Figures and Tables

**Figure 1 ijerph-17-03549-f001:**
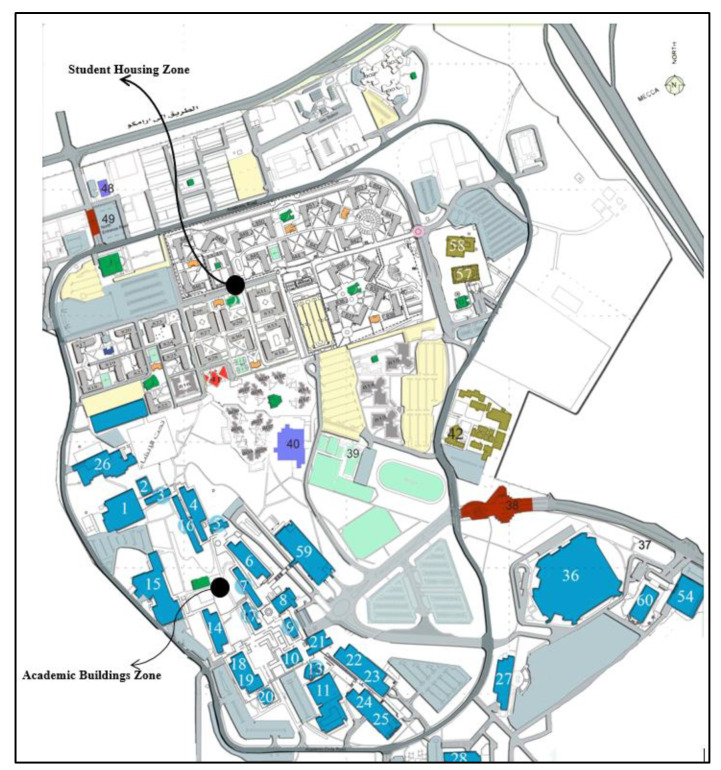
Map of the study area.

**Figure 2 ijerph-17-03549-f002:**
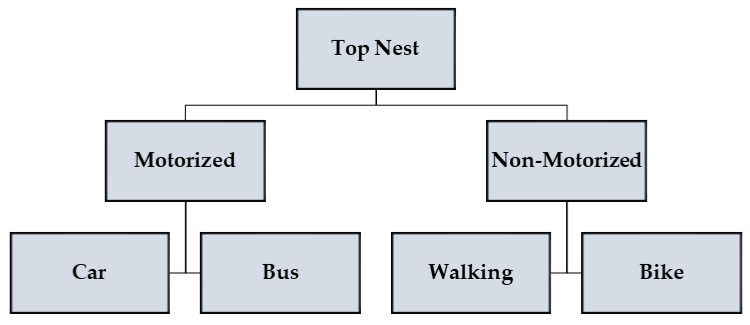
Nest ensemble artificial neural network (ANN).

**Figure 3 ijerph-17-03549-f003:**
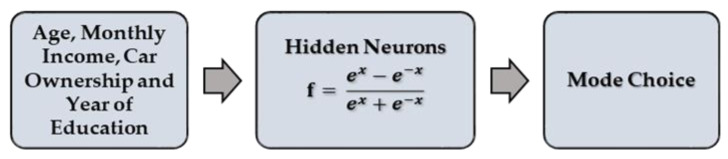
Typical ANN structure.

**Figure 4 ijerph-17-03549-f004:**
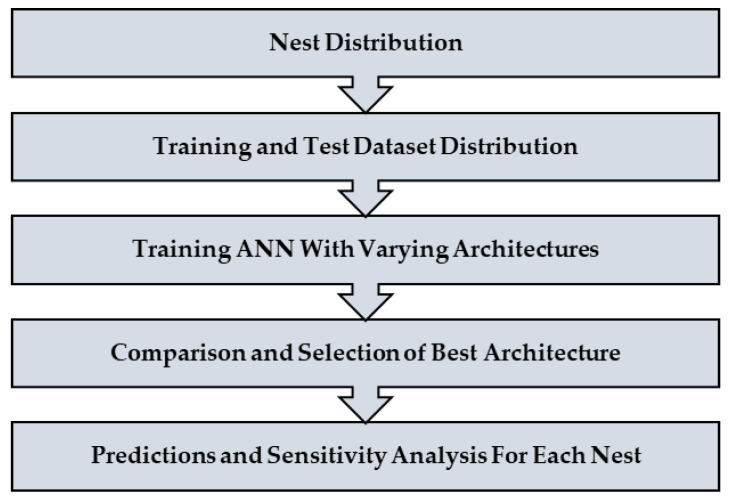
Flowchart for the model development and data analysis process.

**Figure 5 ijerph-17-03549-f005:**
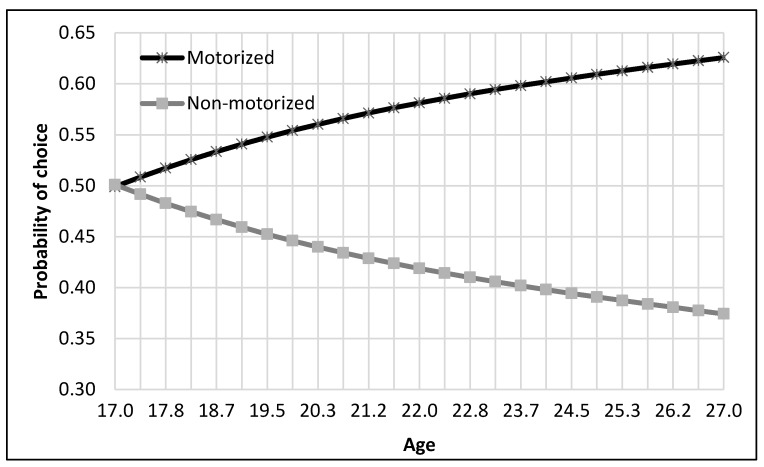
Sensitivity for age for top nest.

**Figure 6 ijerph-17-03549-f006:**
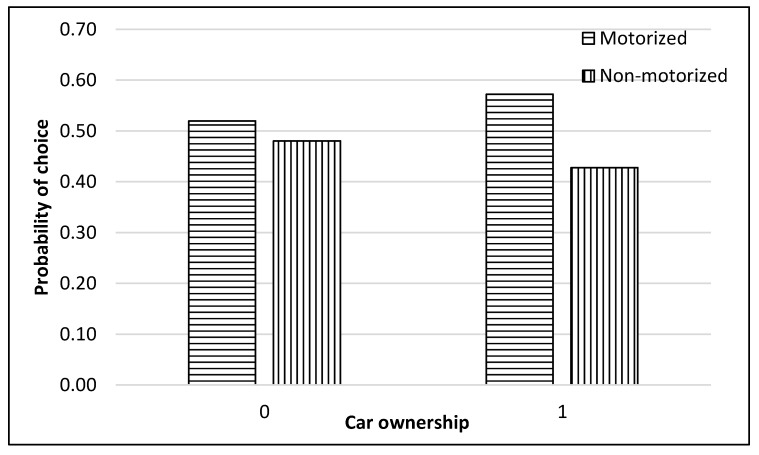
Sensitivity for car ownership for top nNest.

**Figure 7 ijerph-17-03549-f007:**
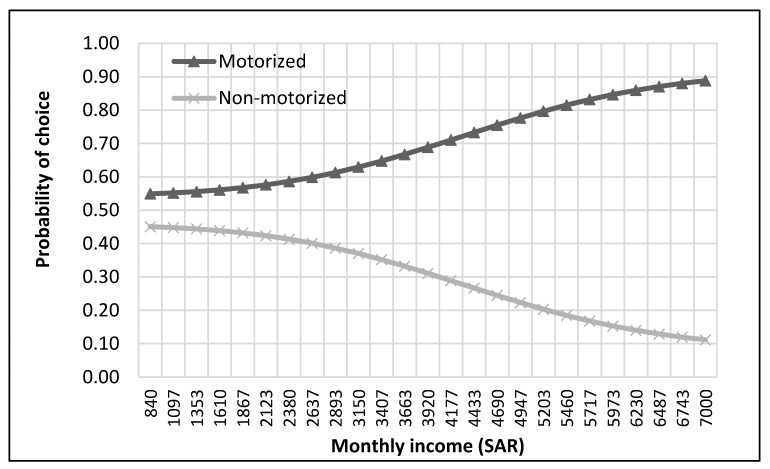
Sensitivity for monthly income for top nest.

**Figure 8 ijerph-17-03549-f008:**
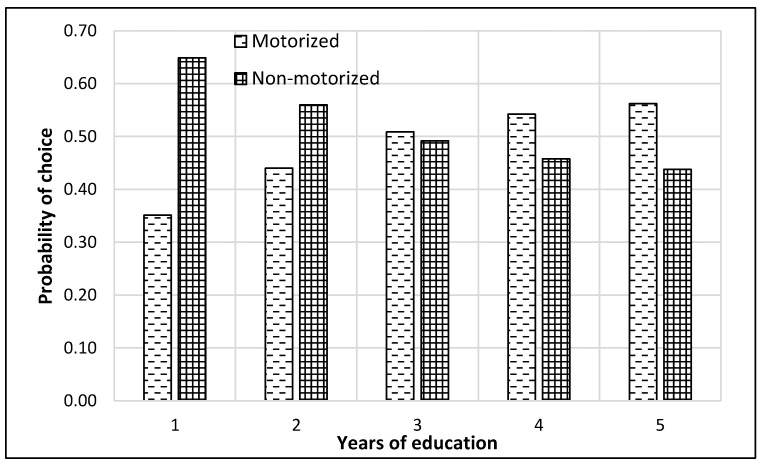
Sensitivity for years of education for top nest.

**Figure 9 ijerph-17-03549-f009:**
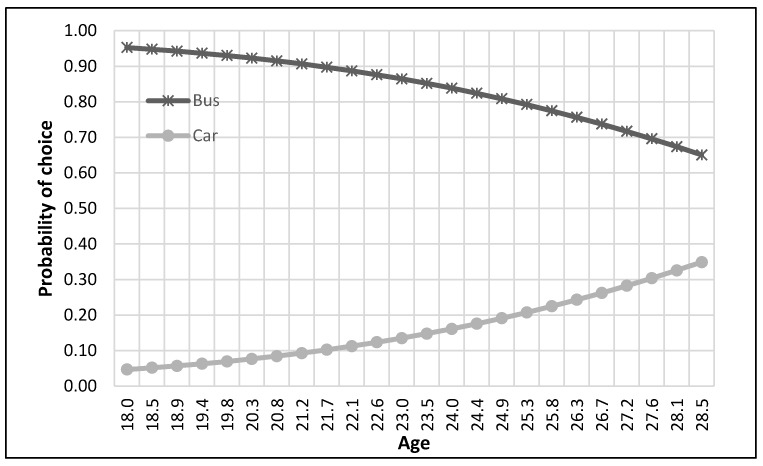
Sensitivity for age for motorized nest.

**Figure 10 ijerph-17-03549-f010:**
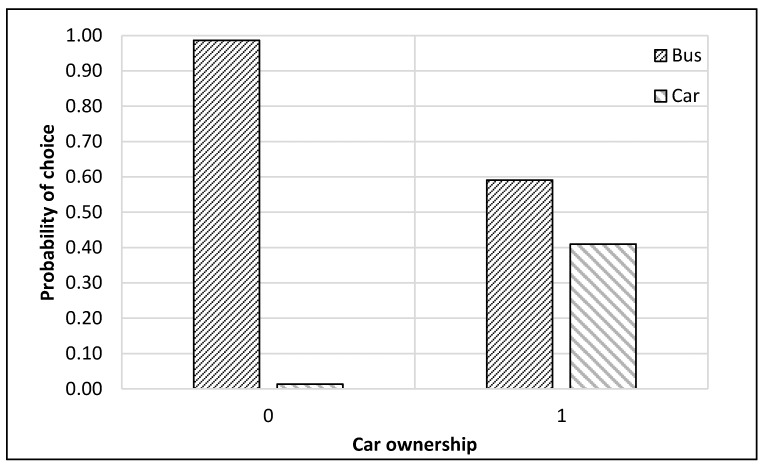
Sensitivity for car ownership for motorized nest.

**Figure 11 ijerph-17-03549-f011:**
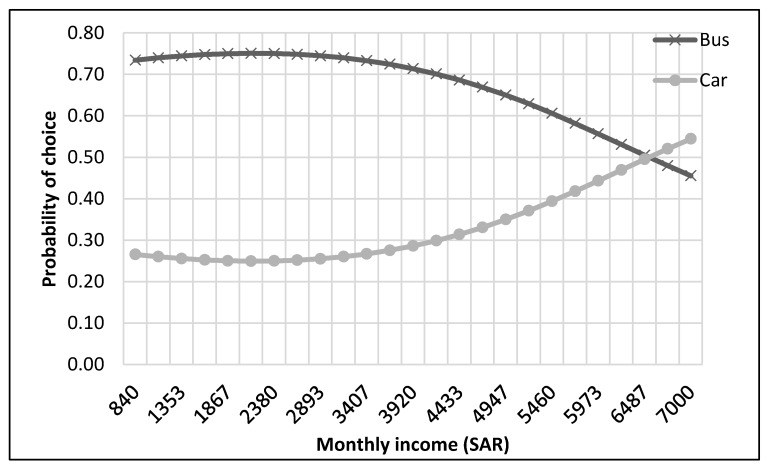
Sensitivity for monthly income for motorized nest.

**Figure 12 ijerph-17-03549-f012:**
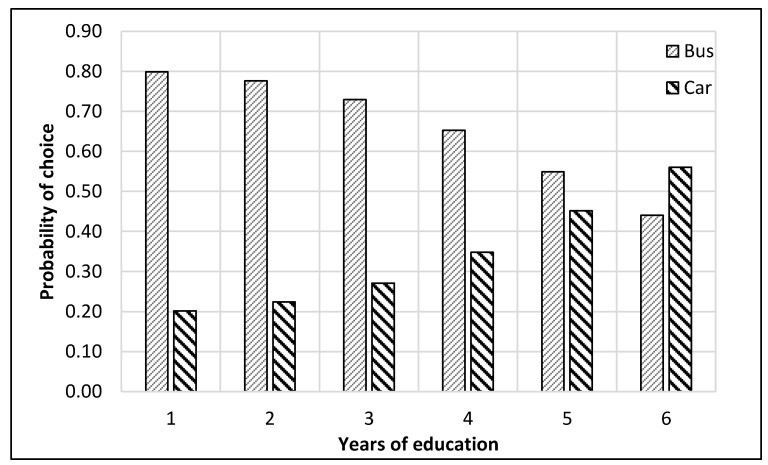
Sensitivity for years of education for motorized nest.

**Figure 13 ijerph-17-03549-f013:**
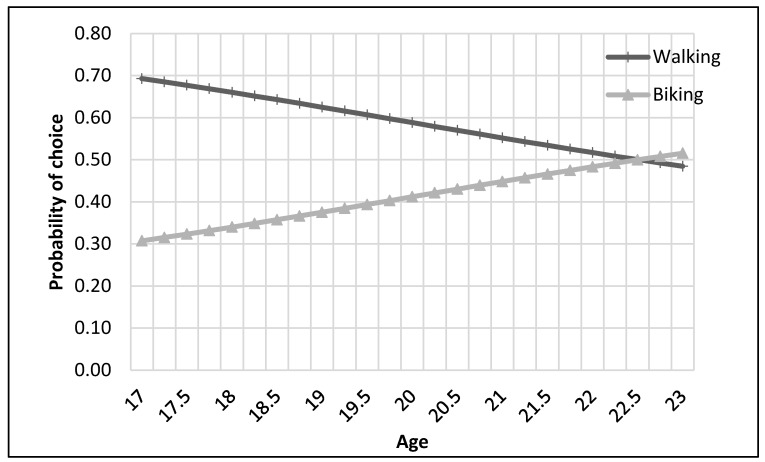
Sensitivity for age for non-motorized nest.

**Figure 14 ijerph-17-03549-f014:**
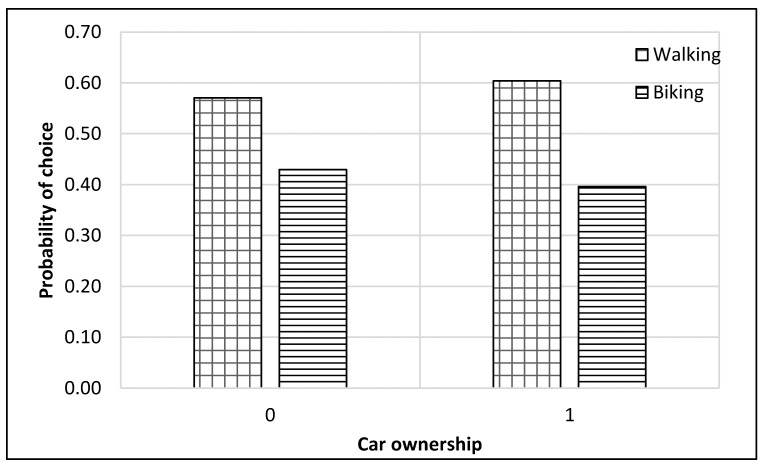
Sensitivity for car ownership for non-motorized nest.

**Figure 15 ijerph-17-03549-f015:**
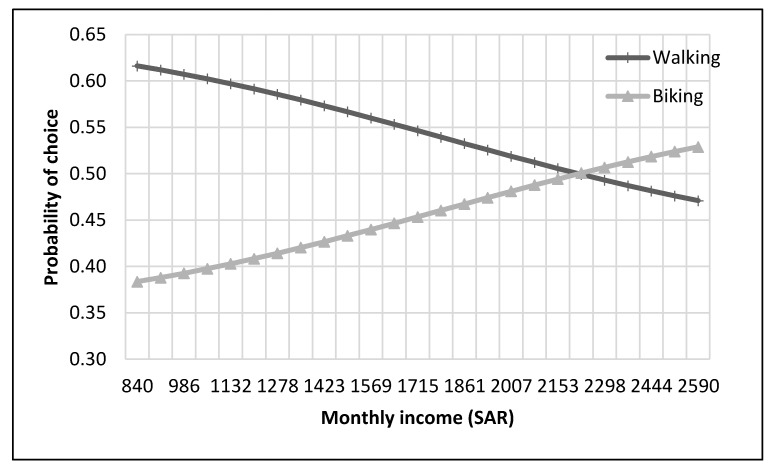
Sensitivity for monthly income for non-motorized nest.

**Figure 16 ijerph-17-03549-f016:**
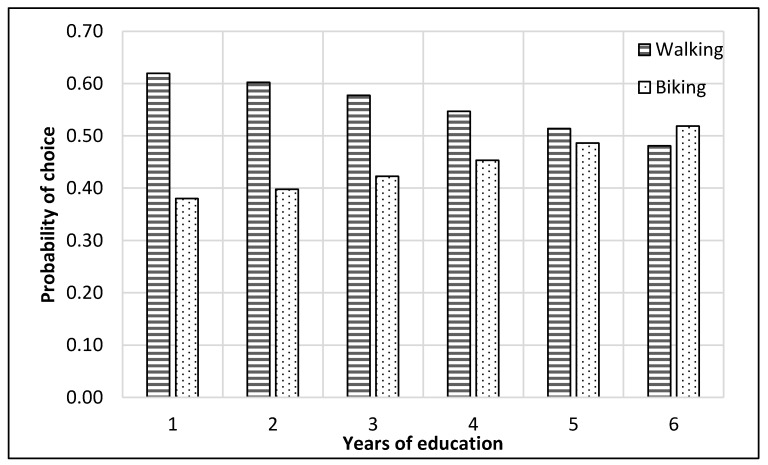
Sensitivity for years of education for non-motorized nest.

**Table 1 ijerph-17-03549-t001:** Transportation impacts on sustainability [10].

Economic	Social	Environmental
Traffic congestion	Public health impacts	Air and water pollution
Damages due to accidents	Community interactions	Loss of habitats
Restricted mobility	Livability	Reduction of non-renewable resources
Cost of facilities	Social inequity	
Reduction of non-renewable resources	Aesthetics	

**Table 2 ijerph-17-03549-t002:** Dataset description.

Variable	Description	
Age	Continuous variable in years	
Income	Continuous variable in SAR	
Car ownership	Binary variable; 1 if the student has a car, otherwise 0	
Education level	Ordinal variable	
	1	Student is in the first or second year (orientation and freshmen) of undergraduate studies
	2	Student is in the third year (sophomore) of undergraduate studies
	3	Student is in the fourth year (junior) of undergraduate studies
	4	Student is in the fifth year (senior) of undergraduate studies
	5	Student is completing a master’s degree
	6	Student is completing a Ph.D.

**Table 3 ijerph-17-03549-t003:** Accuracy of ANN models.

Parameter	Value
Top Nest ANN	
Training accuracy	63%
Test accuracy	60%
Motorized Nest	
Training accuracy	68%
Test accuracy	67%
Non-Motorized Nest	
Training accuracy	68%
Test accuracy	60%

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
