# Peer review of "A Nested Ensemble Approach with ANNs to Investigate the Effect of Socioeconomic Attributes on Active Commuting of University Students"

_ijerph, 2020, doi:10.3390/ijerph17103549_

Round 1
Reviewer 1 Report
This is a highly interesting piece of work. Some improvements can be made based on the comments below.
- The manuscript's title seems to highlight the use of ANN. There should be a clear justification why such a method is used in this study, as opposed to other methods. In short, why was the method used?
- Confining the independent variables to age, monthly income, car ownership, and years of education sounds rather problematic. There are more factors that affect people's mode choice. Further review of literature is required, and additional variables must be included for a more reliable outcome.
- Based on what this study presents as major finding, it is difficult to suggest providing infrastructure as an alternative. There is no evidence offered in the results that installing such infrastructure would change people's behavior.
Author Response
Point 1: The manuscript's title seems to highlight the use of ANN. There should be a clear justification why such a method is used in this study, as opposed to other methods. In short, why was the method used?
Response 1: The main reason for using ANNs in this study is the fact that the utilization of ANNs and other non-parametric methods have been found promising and superior to traditional parametric techniques, see reference [33] and [34]. However, the literature review has shown that they are still under-utilized, especially for nested choice problems. Hence, it was decided by the research team to utilize ANN with a unique approach to incorporate a nested pool of choices.
The last paragraph of section 2 has been modified with the following text to address this comment.
“However, the successful implementation of MLTs, as suggested by the results of this study, is taken as a motivation to use MLTs for university goers’ mode choice with more input variables in this study. Each model has its own set of advantages over others and is also used based on the type of problems presented. For example, in discrete choice models, the number of alternatives is confined. However, in the ordinary regression model, the dependent variable can take any infinite number of values. Multinomial logit (ML) relies on the assumption of independence of irrelevant alternatives (IIA), which is not always desirable. For example, in the ML model the relative probabilities of taking a car or bus to university do not change if a bicycle or other mode of transportation is added as an additional possibility. Thus, often times ML models impose too much constraints on the relative preferences between the different alternatives. Nested logit or the Multinomial Probit model could be used to solve this problem, as they allow for violation of the IIA. With all these compulsion in mind we found that the juxtapose use of MLTs along with nested ensemble has not been successfully used for mode choice problems in literature, which has resulted in its non-utilization in mode choice problems with pool of similar choices. Hence, this study attempts to fulfill this gap.”
Point 2: Confining the independent variables to age, monthly income, car ownership, and years of education sounds rather problematic. There are more factors that affect people's mode choice. Further review of literature is required, and additional variables must be included for a more reliable outcome.
Response 2: In practice, identifying all factors affecting individual choice decisions involve some inaccuracies as their determinants are partially observed or imperfectly measured. Therefore, discrete choice models rely on stochastic assumptions and specifications to account for unobserved factors related to choice alternatives. Thus the set of variables was chosen based on the maximum number of appearances in the literature. While the authors agree with the reviewer that limiting the variables only to age, monthly income, car ownership, and years of education can be tricky in producing the desired outcome; however, this can be taken as one of the limitations of the study.
The following statement has been added to the first paragraph of section 4 to point out this limitation.
“The study was limited to these variables since their relation to mode choice has been observed in the literature [8, 22, 31, 36].”
Point 3: Based on what this study presents as major finding, it is difficult to suggest providing infrastructure as an alternative. There is no evidence offered in the results that installing such infrastructure would change people's behaviour.
Response 3: It was found that motorized modes are generally preferred, and among them, cars are more preferred than buses. We agree with the reviewer that the models did not include the infrastructural elements. Hence, we suggested some general recommendations which have proven useful by other studies in promoting active modes of transportation.
The recommendations in section 7 have been modified as follows:
“The study’s findings reveal that motorized modes (especially cars) are more preferred by the students. Hence, there is a need to take measures to promote non-motorized modes of transport among young students, especially those with higher income levels. It has been found from the review of literature that this could be accomplished by providing favorable infrastructure for these active modes; this includes providing shaded pathways, bike lanes, and parking spaces. Moreover, providing insight into the benefits of commuting via bicycles to orientation students and first-year students could motivate them to use bikes at an early stage of their college career. Operating a shuttle service in the evenings and connecting it to nearby off-campus locations also might encourage graduate students using motorized modes, who are presently biased to commute by cars, to use the service. However, the effects of these measures could be studied in detail if and when they are implemented.”
Reviewer 2 Report
This paper presents an analysis on the effect of socioeconomic attributes on commute mode choice of university students.
My main concern with the paper is the focus and relevance. What exactly does this study add? Why the focus on the effect of socioeconomics and not for example the effect of available infrastructure? What kind of recommendations can follow from this analysis?
The introduction seems unfinished. It ends with a Table, but it is not clear what this Table adds to the story. The introduction lacks a clear research gap and aim of the study. (it does not present the structure of the paper either)
The end of the literature study mentions a number of studies and their analysis methods, but they are not explained. E.g. discrete choice models, MNL models, nested models, econometric (nested logit) approach are mentioned as if they are all different, but it is not properly explained why a certain method is to be preferred over another.
In section 5, it does not really become clear what exact model is used.
The results section does not present a clear overview of the full model, including goodness of fit or significance of parameters.
The results indicate that older age is related with a larger likelihood of motorized modes, while more years of education is related with a smaller likelihood of motorized modes. Are age and years of education not strongly correlated? So how can this difference be explained? Multicollinearity?
However, as indicated, my main issue with the paper is the question why this is relevant and what it adds to the existing knowledge.
THe conclusion stateds that the findings reveil the need to make policies to promote non-motorized modes and this could be accomplished by providing shaded pathways, bike lanes, ect. However, the analyses are not related to infrastructure, so what do the analyses add to underpin these recommendations?
Author Response
Point 1: My main concern with the paper is the focus and relevance. What exactly does this study add? Why the focus on the effect of socioeconomics and not for example the effect of available infrastructure? What kind of recommendations can follow from this analysis?
Point 2: The introduction seems unfinished. It ends with a Table, but it is not clear what this Table adds to the story. The introduction lacks a clear research gap and aim of the study. (It does not present the structure of the paper either).
Response 1-2 : The authors thought that the infrastructure would have no effect on the mode choice as all respondents are students in KFUPM which means that they have same infrastructure.
The following paragraphs about the scope and relevance has been added at the end of section 1:
“Active commuting by university students might appear to have limited effects on society, as it occurs only at the university campus. However, it could be considered a stepping stone for achieving a sustainable society. Only a limited number of studies have been conducted on university populations, not including any travel behavior study on a university population in the context of Saudi Arabia. Thus, this study aims to explore the effect of socioeconomic attributes on active traveling at the university level. A nested ensemble approach with artificial neural networks (ANNs) has been used to assess the effect of socioeconomic factors on students’ active commuting decisions. The uniqueness of this ANNs-based modeling approach used for this research is unprecedented in this field of study. It is expected that the results of the study can be utilized to identify the influencing factors for active commuting for university populations throughout Saudi Arabia. These factors can be considered useful for regional travel demand modeling. Also, this study utilizes a rich data set on the activity-travel behavior of a large Saudi Arabian university population, which can be considered a valuable addition to the literature. Knowledge of critical factors influencing the choice for such active modes might serve as a valuable input parameter for university policymakers.
The paper is organized as follows: Section 2 contains a review of the relevant literature. Section 3 describes the study area. The data collection method used in the study is then described in section 3, after which the methodology is presented in section 4. Results are presented and discussed in section 5. Finally, section 6 outlines the main conclusions and identifies and recommendations for further research.”
Regarding the recommendations, The recommendations at the end of section 7 have been modified as follows:
“The study’s findings reveal that motorized modes (especially cars) are more preferred by the students. Hence, there is a need to take measures to promote non-motorized modes of transport among young students, especially those with higher income levels. It has been found from the review of literature that this could be accomplished by providing favorable infrastructure for these active modes; this includes providing shaded pathways, bike lanes, and parking spaces. Moreover, providing insight into the benefits of commuting via bicycles to orientation students and first-year students could motivate them to use bikes at an early stage of their college career. Operating a shuttle service in the evenings and connecting it to nearby off-campus locations also might encourage graduate students using motorized modes, who are presently biased to commute by cars, to use the service. However, the effects of these measures could be studied in detail if and when they are implemented.”
Point 3: The end of the literature study mentions a number of studies and their analysis methods, but they are not explained. E.g. discrete choice models, MNL models, nested models, econometric (nested logit) approach are mentioned as if they are all different, but it is not properly explained why a certain method is to be preferred over another.
Response 3: The text in the last paragraph of section 2 has been modified as follows:
Each model has its own set of advantages over others and are also used based on the type of problems presented, For example, in discrete choice models the number of alternatives is confined where as in ordinary regression model the dependent variable can take any infinite number of values. Multinomial Logit (ML) relies on the assumption of independence of irrelevant alternatives (IIA), which is not always desirable. For example, in ML model the relative probabilities of taking a car or bus to university do not change if a bicycle or other mode of transportation is added as an additional possibility. Thus often times ML models impose too much constraints on the relative preferences between the different alternatives. Nested logit or the Multinomial Probit model could be used to solve this problem, as they allow for violation of the IIA. With all these compulsion in mind we found that the juxtapose use of MLTs along with nested ensemble has not been successfully used for mode choice problems in literature, which has resulted in its non-utilization in mode choice problems with pool of similar choices. Hence, this study attempts to fulfill this gap.
Point 4: In section 5, it does not really become clear what exact model is used.
Response 4: A series of ANNs were used for the prediction of mode choice in this study. Each ANN was used to predict the mode choice of the traveler from a specified set (nest). So, the first step was the prediction of traveler to go for active mode choices (non-motorized) or motorized choices. Then for each of these sets, an ANN was developed to predict the final choice of the traveler. Hence, the ultimate choice of the traveller was the result of predictions from two ANNs, which is why it is called an ensemble.
The text in the first paragraph of section 5 is modified as follows:
“Students’ mode choice was predicted using a series of ANN models, depicted in Figure 1. For this approach, the final mode choice is determined in two steps; the ANN (top nest) was to predict the type of mode (motorized and non-motorized), while the actual mode choice was predicted in the last step, with another ANN, depending on the category of mode (motorized or non-motorized) the traveler has selected in the first step. Therefore, each classification of mode choice is referred to as a nest—since the final mode choice is an outcome of two ANNs—therefore it is referred to as an ensemble.”
Point 5: The results section does not present a clear overview of the full model, including goodness of fit or significance of parameters.
Response 5: ANNs are non-parametric artificial intelligence models. Unlike statistical models, such as logit, ANNs do not have restrictions for the statistical significance of parameters or the models themselves. Hence, their goodness of fit is represented by their accuracies in table 3.
Point 6: The results indicate that older age is related with a larger likelihood of motorized modes, while more years of education is related with a smaller likelihood of motorized modes. Are age and years of education not strongly correlated? So how can this difference be explained? Multicollinearity?
Response 6: we thank the reviewer for pointing out this issue. The age and education should be correlated to some extent. However, it may not be accurate for graduate students, and this was the reason that these variables were kept in the model. The general trend is that older students are more likely to use motorized modes, especially cars. On rechecking the results, some issues related to coding of categorical variables in plotting some figures were found, which have been corrected in the updated manuscript.
Figures 8 and 9 have been corrected, and the corresponding text at the end of section 6.1 and start of section 6.2 has been modified as follows:
“It should be noted at this point that many of these graduate students are working/on-scholarship; therefore, they would find it easier to own and maintain a car. It is consistent with the observations from Figure 5.”
“However, age negatively impacts the probability of bus use, as displayed in Figure 9.”
Point 7: However, as indicated, my main issue with the paper is the question why this is relevant and what it adds to the existing knowledge.
Response 7: The following paragraphs have been added at the end of section 1.
“Active commuting by university students might appear to have limited effects on society, as it occurs only at the university campus. However, it could be considered a stepping stone for achieving a sustainable society. Only a limited number of studies have been conducted on university populations, not including any travel behavior study on a university population in the context of Saudi Arabia. Thus, this study aims to explore the effect of socioeconomic attributes on active traveling at the university level. A nested ensemble approach with artificial neural networks (ANNs) has been used to assess the effect of socioeconomic factors on students’ active commuting decisions. The uniqueness of this ANNs-based modeling approach used for this research is unprecedented in this field of study. It is expected that the results of the study can be utilized to identify the influencing factors for active commuting for university populations throughout Saudi Arabia. These factors can be considered useful for regional travel demand modeling. Also, this study utilizes a rich data set on the activity-travel behavior of a large Saudi Arabian university population, which can be considered a valuable addition to the literature. Knowledge of critical factors influencing the choice for such active modes might serve as a valuable input parameter for university policymakers.”
Point 8: The conclusion stated that the findings reveal the need to make policies to promote non-motorized modes and this could be accomplished by providing shaded pathways, bike lanes, ect. However, the analyses are not related to infrastructure, so what do the analyses add to underpin these recommendations?
Response 8: It was found that motorized modes are generally preferred, and among them, cars are more preferred than buses. We agree with the reviewer that the models did not include the infrastructural elements. Hence, we suggested some general recommendations which have proven useful by other studies in promoting active modes of transportation.
The recommendations have been modified as follows:
“The study’s findings reveal that motorized modes (especially cars) are more preferred by the students. Hence, there is a need to take measures to promote non-motorized modes of transport among young students, especially those with higher income levels. It has been found from the review of literature that this could be accomplished by providing favorable infrastructure for these active modes; this includes providing shaded pathways, bike lanes, and parking spaces. Moreover, providing insight into the benefits of commuting via bicycles to orientation students and first-year students could motivate them to use bikes at an early stage of their college career. Operating a shuttle service in the evenings and connecting it to nearby off-campus locations also might encourage graduate students using motorized modes, who are presently biased to commute by cars, to use the service. However, the effects of these measures could be studied in detail if and when they are implemented.”
Reviewer 3 Report
Dear authors,
the paper is well organized and the research is explained in a clear a deep way. I think that the paper could be considered ready for publication.
Anyway, I would like to suggest a small improvement regarding:
- a clearer explanation of the research’s aim and scope at the beginning of the paper;
- since the topic and the implications for the community are so relevant, I think that more detailed consideration about the research’s next steps or about further works that this research is opening, could be interesting (line 354-358);
- moreover, a suggestion could be to add the results of the questionnaires (229) as added table at the end of the paper, as appendix.
As previously said, regarding the structure of the paper, introduction and literature review, methodology and presentation of results, as well as conclusions, I think that those issues are well presented.
Best regards
Author Response
Point 1: A clearer explanation of the research’s aim and scope at the beginning of the paper.
Response 1: The following paragraph has been added at the end of section 1.
“Active commuting by university students might appear to have limited effects on society, as it occurs only at the university campus. However, it could be considered a stepping stone for achieving a sustainable society. Only a limited number of studies have been conducted on university populations, not including any travel behavior study on a university population in the context of Saudi Arabia. Thus, this study aims to explore the effect of socioeconomic attributes on active traveling at the university level. A nested ensemble approach with artificial neural networks (ANNs) has been used to assess the effect of socioeconomic factors on students’ active commuting decisions. The uniqueness of this ANNs-based modeling approach used for this research is unprecedented in this field of study. It is expected that the results of the study can be utilized to identify the influencing factors for active commuting for university populations throughout Saudi Arabia. These factors can be considered useful for regional travel demand modeling. Also, this study utilizes a rich data set on the activity-travel behavior of a large Saudi Arabian university population, which can be considered a valuable addition to the literature. Knowledge of critical factors influencing the choice for such active modes might serve as a valuable input parameter for university policymakers.”
Point 2: since the topic and the implications for the community are so relevant, I think that more detailed consideration about the research’s next steps or about further works that this research is opening, could be interesting (line 354-358);
Response 2: The authors would like to thank the reviewer for his encouraging comment.
Point 3: Moreover, a suggestion could be to add the results of the questionnaires (229) as added table at the end of the paper, as appendix.
Response 3: Table A1 has been added under Appendix A.
Reviewer 4 Report
Understanding the concept of sustainable transportation is essential to ensure a clean, healthy, and high quality of the environment. The sustainable transportation concept also emphasizes human life and the environment to meet current and future needs, without considering the ability of the future generation to meet its own needs. The manuscript “A Nested Ensemble Approach with ANNs to Investigate the Effect of Socioeconomic Attributes on Active Commuting of University Students” (ID: ijerph-779145) investigate the effects of socioeconomic characteristics on the mode choice of university students. The selected topic has high theoretical significance and academic value. This research strengthens the practical understanding of the compact motorized modes, and can support the design and planning of motorized modes. The English language of the manuscript is well written. However, there are some scientific problems in the current status of this manuscript. The detailed comments and suggestions are listed as follows:
1) Line 67: Table 1 should provide references to various influencing factors.
2) Line 160: It is suggested to supplement the general map of the study area and the road distribution.
3) Line 163: We recommend that relevant data sources be supplemented.
4) Line 180: The authors should explain the rationality of the choice of 246 questionnaires based on the actual population of the study area.
5) Line 183: The authors should provide references for different sources of influencing factors.
6) Line 184: The author should test the validity of the questionnaire.
7) Line 197: We suggest that the author beautify the picture in the paper.
8) Line 214: The author should specify the implementation of the artificial neural network, the programming software and software package used. It is suggested to add flowchart for illustration.
9) Line 240: The author should give the sample size of the model construction and the sample size of the precision test. In addition, the author should also give the selection method of the precision test sample, such as random sampling.
10) Line 255: The authors should specify the method of sensitivity analysis in the data analysis section.
In the end, I recommend minor revision the manuscript.
Author Response
Point 1: Line 67: Table 1 should provide references to various influencing factors.
Response 1: The following Reference has been cited at proper place with the caption of the table.
Ref #10 Litman, T. and D. Burwell, Issues in sustainable transportation. International Journal of Global Environmental Issues, 2006. 6(4): p. 331-347.
Point 2: Line 160: It is suggested to supplement the general map of the study area and the road distribution.
Response 2: Figure 1 has been added under study area section.
Point 3: Line 163: We recommend that relevant data sources be supplemented.
Response 3: It should be mentioned that the data used in developing ANNs was used as a first time in this study and it was collected through an online questionnaire survey as mentioned in section 4. The authors are ready to provide their dataset as supplementary material if required. Moreover, the following reference about climate conditions during summer in the study area has been added under section 3.
Ref# 35 The General Authority of Meteorology & Environmental Protection, S.A., Summer Climate Report for Saudi Arabia. 2019: Riyadh.
Point 4: Line 180: The authors should explain the rationality of the choice of 246 questionnaires based on the actual population of the study area.
Response 4: It should be noted that the number “246” represents the total number of responses received. As mentioned in Data Collection and Description section, the questionnaire was online and it was distributed using different means such as twitter hashtag, online forums and through the university student affairs office. The total number of valid responses found to be 229 responses which represents an acceptable sample size for study areas with few subgroups such as KFUPM.
The following two references about the sample size have added under section 4:
Ref # 38 MacCallum, R. C., Widaman, K. F., Zhang, S., & Hong, S. (1999). Sample size in factor analysis. Psychological methods, 4(1), 84.
Ref # 39 . Sudman, S. (1976). Applied sampling, New York: Academic Press.
Point 5: Line 183: The authors should provide references for different sources of influencing factors.
Response 5: The following references have been cited under section 4 to address the reviewer comment:
Ref #8. Delmelle, E.M. and E.C. Delmelle, Exploring spatio-temporal commuting patterns in a university environment. Transport Policy, 2012. 21: p. 1-9.
Ref #22. Daisy, N.S., et al., Understanding and modeling the activity-travel behavior of university commuters at a large Canadian university. Journal of Urban Planning and Development, 2018. 144(2): p. 04018006.
Ref #3 Zhou, J., Y. Wang, and J. Wu, Mode choice of commuter students in a college town: an exploratory study from the United States. Sustainability, 2018. 10(9): p. 3316.
Ref #36. Vij, A., A. Carrel, and J.L. Walker, Incorporating the influence of latent modal preferences on travel mode choice behavior. Transportation Research Part A: Policy and Practice, 2013. 54: p. 164-178.
Point 6: Line 184: The author should test the validity of the questionnaire.
Response 6: The validity of the questionnaire was checked by conducting a participatory pilot survey before the full-scale survey. The questionnaire was administered to a small sample of students who were told that they are in the pre-test phase and they were asked about their comments and suggestions on the questionnaire. Finally, all issues discovered during the pilot survey phase were addressed before executing the large-scale survey phase. The following paragrapgh has been added to the first paragrapgh of section 4
“A participatory pilot study was conducted before the full-scale survey to ensure that the questions and answers of the questionnaire are meaningful [37]. The questionnaire was administered to a small sample of students. The students were interviewed and told that they are in the pre-test phase. Then, they were asked about their comments and suggestions on the questionnaire. All issues discovered during the pilot survey phase were addressed. The responses to the questionnaire were interpolated to develop a modified questionnaire that has been used for this study.”
Point 7: Line 197: We suggest that the author beautify the picture in the paper.
Response 7: We thank the reviewer for this comment. The shape of Figures 2 & 3 have been updated.
Point 8: Line 214: The author should specify the implementation of the artificial neural network, the programming software and software package used. It is suggested to add flowchart for illustration.
Point 9: Line 240: The author should give the sample size of the model construction and the sample size of the precision test. In addition, the author should also give the selection method of the precision test sample, such as random sampling.
Point 10: Line 255: The authors should specify the method of sensitivity analysis in the data analysis section.
Response 8-10: ANNs were developed using Statistica program. Firstly, data was distributed as per mode choices in to different nests. Then, random sampling was applied to distribute the data in to training and test datasets which consisted of 75% and 25% of the sample size respectively. The number of samples for each ANN model were as follows; 172 for training and 57 for testing ANN of motorized and non-motorized mode choices, 106 for training and 35 for testing ANN of bus and car choices, and 66 for training and 22 for testing walking and bike choices. Each ANN was trained with different number of hidden neurons and their accuracies were compared with variation of hidden neurons and optimum number of neurons were selected on the basis of best accuracy in each case. Sensitivity analysis was performed by selecting one variable at a time and then the values of that variable were changed in the model and mode choice probabilities were calculated using the model. Other variables of the model were kept constant in this process. The change in mode choice probabilities to see the effect of change in a specific variable on them.
Figure 3 has been added. Last paragraphs of section 5 (Modeling Methodology) have been modified as follows:
“The complete process of modeling and analysis is shown in Figure 4. It starts with the distribution of data into different nests according to the mode choice of travelers (see Figure 2). Then each dataset was divided, through randomly sampling, into training and testing datasets with 75% samples for training and 25% for testing. The dataset consisted of 229 samples. Thus, the training dataset was 172 samples, while the test dataset was 57 samples for the top nest ANN model. There were 141 samples for the motorized mode choices, among which 106 were training samples, and 35 were test samples for the ANN model developed for motorized mode choice nest. Lastly, 88 samples were available for non-motorized mode choice nest, and 66 were for training, and 22 were the test of ANN model for non-motorized mode choices. ANNs were developed using Statistica (by Statsoft Inc.) Then, their accuracies were compared to find the optimum number of neurons. Finally, a sensitivity analysis was performed by selecting one variable, at a time, and changing its value from the given range in the data sample. The model was used to predict the values of the mode choice of students for each ANN by changing the value of one variable while other variables were not changed in that case. For example, to perform sensitivity analysis for age, only its value was changed while other values were constant. The probabilities of mode choices, predicted by the model with the change in the values of a variable, were then plotted (see Figures 4-15) to show the effect of that variable on them. In each case, the model was used to perform a sensitivity analysis by changing the values of specific variables while keeping other variables at a constant value (average of the variable). The range of parameters was taken from the actual data; for example, the income of students ranges from approximately SAR 800 ($213) to SAR 7000 ($1,867), as it was the range of income for the top nest (see Figure 7 ). On the other hand, students using non-motorized modes have a lower income range (SAR 840–SAR 2590).”
Round 2
Reviewer 1 Report
Significant improvements and clarifications are made in this revision. I recommend publication.
Reviewer 2 Report
I am still concerned about the limited contribution of the paper. It only focuses on the relationship between socio-demographics and mode choice of students based on a sample of students of one campus. Therefore, the recommendations that follow from this study are also limited.
The study has been properly conducted and described.
Author Response
We acknowledge the concern of the reviewer. We would like to notify the following related to contribution of the paper. This is the first study related to mode choice of university students for Saudi Arabia. King Fahd University is a leading university in Saudi Arabia and has a wide variety of students, hence, the findings of this study are expected to be applicable in a wider perspective as well. Lastly, one of the major contributions of this paper is the use of ANN nested ensembles which has not been found in any literature related to mode choice modeling.